# Domain adaptation through anatomical constraints for 3d human pose estimation under the cover

**Alexander Bigalke**[1]                              ALEXANDER.BIGALKE@UNI-LUEBECK.DE
**Lasse Hansen**[1]                                              L.HANSEN@UNI-LUEBECK.DE
**Jasper Diesel**[2]                                          JASPER.DIESEL@DRAEGER.COM
**Mattias P. Heinrich**[1]                              MATTIAS.HEINRICH@UNI-LUEBECK.DE

[1] *Institute of Medical Informatics, University of Lübeck*
[2] *Drägerwerk AG & Co. KGaA*

**Editors:** Under Review for MIDL 2022

## Abstract

Domain adaptation has the potential to overcome the expensive or even infeasible labeling of target data by transferring knowledge from a labeled source domain. In this work, we address domain adaptation in the context of point cloud-based 3D human pose estimation, whose clinical applicability is severely limited by a lack of labeled training data. Unlike the mainstream approach of domain-invariant feature learning, we propose to guide the learning process in the target domain through weak supervision, based on prior knowledge about human anatomy. We embed this prior knowledge into a novel loss function that encourages network predictions to match the statistics of an anatomically plausible skeleton. Specifically, we formulate three loss functions that penalize asymmetric limb lengths, implausible joint angles, and implausible bone lengths. We evaluate the method on a public lying pose dataset (SLP), adapting from uncovered patients in the source to covered patients in the target domain. Our method outperforms diverse state-of-the-art domain adaptation techniques and improves the baseline model by 26 % while reducing the gap to a fully supervised model by 54 %. Source code is available at https://github.com/multimodallearning/da-3dhpe-anatomy.

**Keywords:** Domain adaptation, 3D pose estimation, anatomical constraints, point clouds

## 1. Introduction

3D human pose estimation has diverse clinical applications, such as patient monitoring (Chen et al., 2018) or context-aware assistance systems in the operating room (Hansen et al., 2019). As for most other vision tasks, deep learning-based methods have substantially advanced the state of the art for human pose estimation (Chen et al., 2020). Strong performance, however, is closely tied to the availability of large-scale annotated datasets (Ionescu et al., 2013). While the annotation of 3D poses is generally laborious, it is even more problematic in a clinical setting. In the context of patient monitoring, for instance, not only is the privacy of patients to be respected, but occlusions of the patients by a blanket make accurate annotations nearly impossible. We address the first of the two issues by using point cloud data, which is not only anonymity-preserving (Silas et al., 2015) but also constitutes a natural modality for 3D pose estimation as it inherently preserves the 3D structure of the scene. Regarding the second issue, the focus of this work, domain adaptation (Wang and Deng, 2018) has the potential to overcome the lack of labeled target data by adapting

a model from a source domain where rich annotations are available. Altogether, efficient domain adaptation for point cloud-based 3D human pose estimation is thus an important task to advance clinical monitoring systems.

A popular approach for domain adaptation couples supervised task learning in the source domain with the learning of domain-invariant source and target features, realized by discrepancy minimization (Tzeng et al., 2014), adversarial learning (Ganin and Lempitsky, 2015; Tzeng et al., 2017), or reconstruction (Bousmalis et al., 2016; Ghifary et al., 2016). This procedure, however, has the weakness that target features are not optimized for the actual task, and domain invariance does not necessarily induce task relevance (Saito et al., 2018). Further, such approaches usually align global feature vectors, which can be insufficient if solving the task requires the detection of patterns at multiple scales, as is the case for semantic segmentation or 3D pose estimation (Tsai et al., 2018).

Therefore, in the spirit of output space adaptation (Tsai et al., 2018), we aim to supervise the learning process directly in the output space of the target domain. While previous work accomplished this by adversarial learning (Yang et al., 2018), we draw inspiration from recent work on unsupervised pose estimation that embeds prior anatomical knowledge into a deformable shape template (Schmidtke et al., 2021). We also leverage such general domain-independent prior knowledge about human anatomy, but our key idea is to use it as a source of weak supervision in the absence of labels. Specifically, we propose to guide the learning process in the target domain by imposing explicit anatomical constraints on the output space such that network predictions represent anatomically plausible skeletons (Fig. 1). To this end, we formulate three loss terms that penalize asymmetric limb lengths, implausible bone lengths, and implausible joint angles. These losses are jointly minimized with the supervised task loss in the source domain to ensure that predictions are both anatomically plausible and consistent with the observed input. Thus, our method is compatible with arbitrary model architectures and keeps the adaptation procedure simple. It can be optimized by a single forward-backward pass and does not involve adversarial optimization (Yang et al., 2018), multi-step optimization (Saito et al., 2018), or additional network modules (Tzeng et al., 2017; Bousmalis et al., 2016). In summary, the main contributions of this work are:

- We address domain adaptation in the context of 3D human pose estimation by imposing anatomical constraints on the output space of the target domain.

- We formulate three loss functions that penalize asymmetric limb lengths, implausible bone lengths, and implausible joint angles.

- We evaluate the method on the SLP dataset (Liu et al., 2020), adapting from uncovered patients in the source to covered patients in the target domain, and demonstrate that our method is superior to a comprehensive set of state-of-the-art domain adaptation methods, which we adapted to the given problem.

## 2. Related work

Our method is conceptually related to the alignment of output distributions. Tsai et al. (2018) proposed this technique for semantic segmentation, where the distributions of predicted source and target segmentation masks are aligned by training the segmentation

network in an adversarial manner against a discriminator. Yang et al. (2018) and Zhang et al. (2020) introduced a similar idea for 3D human pose estimation and trained a discriminator to differentiate between predicted and ground truth skeletons. In a different approach, applied to keypoint estimation of 3D objects, Zhou et al. (2018) regularize predictions in the target domain by minimizing the Chamfer distance to ground truth labels from the source domain. Unlike these approaches, we implement output space adaptation by embedding explicit anatomical constraints in the loss function. Technically, this is related to the concept of constrained loss functions for medical image segmentation, which was introduced by Kervadec et al. (2019) in the context of weakly supervised learning and transferred to domain adaptation by Bateson et al. (2019). The employed size losses, however, are not applicable to the pose estimation problem, which requires its own constraints and a specifically tailored loss function. Further adaptation techniques that apply supervision in the output space of the target domain include self-ensembling (French et al., 2017) and self-training with pseudo labels (Zou et al., 2018), which were applied to 2D clinician pose estimation (Srivastav et al., 2021) and 2D animal pose estimation (Mu et al., 2020; Li and Lee, 2021), respectively. None of these works incorporate explicit prior knowledge about human anatomy. Explicit anatomical loss functions were used by Sun et al. (2017) and Cao and Zhao (2020) who propose a bone loss and a symmetry loss, respectively. However, they consider a supervised setting, where accurate labels alongside precise bone lengths are known. This differs from the unsupervised setting in our work, which requires the formulation of weaker constraints and a different optimization procedure.

Although not our primary methodological focus, we briefly discuss deep learning on irregular 3D point clouds. The seminal PointNet (Qi et al., 2017a) extracts point-wise spatial embeddings and aggregates them by max-pooling. Various follow-up works proposed hierarchical grouping (Qi et al., 2017b) and generic convolutions (Li et al., 2018; Liu et al., 2019; Wang et al., 2019; Wu et al., 2019; Xu et al., 2021) to incorporate local geometric structure. Among these works, we adapt DGCNN (Wang et al., 2019), based on dynamic graph convolutions, as our point cloud-based pose estimator.

## 3. Methods

We address unsupervised domain adaptation in the context of point cloud-based 3D human pose estimation. Following the classical setting, training data comprises a labeled source dataset $\mathcal{S}$ and an unlabeled target dataset $\mathcal{T}$. The source dataset $\mathcal{S}$ consists of pairs $(\boldsymbol{X}^s, \boldsymbol{Y}^s)$ of 3D point clouds $\boldsymbol{X}^s \in \mathbb{R}^{N \times 3}$ and corresponding labels $\boldsymbol{Y}^s \in \mathbb{R}^{K \times 3}$, which represent the 3D ground truth coordinates of $K$ joints of interest. The target dataset $\mathcal{T}$ contains 3D point clouds $\boldsymbol{X}^t$ without any labels. Given the training data, the goal is to learn a function $f$ with parameters $\boldsymbol{\theta}_f$ that estimates 3D joints as $\hat{\boldsymbol{Y}} = f(\boldsymbol{X}; \boldsymbol{\theta}_f)$ and that achieves the optimal performance on the target domain at test time.

An overview of our proposed method to solve the problem is shown in Fig. 1. We learn the function $f$ by minimizing the joint loss function

$$\mathcal{L}(\boldsymbol{\theta}_f; \mathcal{S}, \mathcal{T}) = \mathcal{L}_{task}(\boldsymbol{\theta}_f; \mathcal{S}) + \lambda \mathcal{L}_{anat}(\boldsymbol{\theta}_f; \mathcal{T}) \tag{1}$$

which is composed of a task loss $\mathcal{L}_{task}$ and an anatomical loss $\mathcal{L}_{anat}$, weighted by the factor $\lambda$. The task loss $\mathcal{L}_{task} = \sum_k \|\hat{\boldsymbol{y}}_k - \boldsymbol{y}_k\|_1 / K$ is implemented as a standard L1-

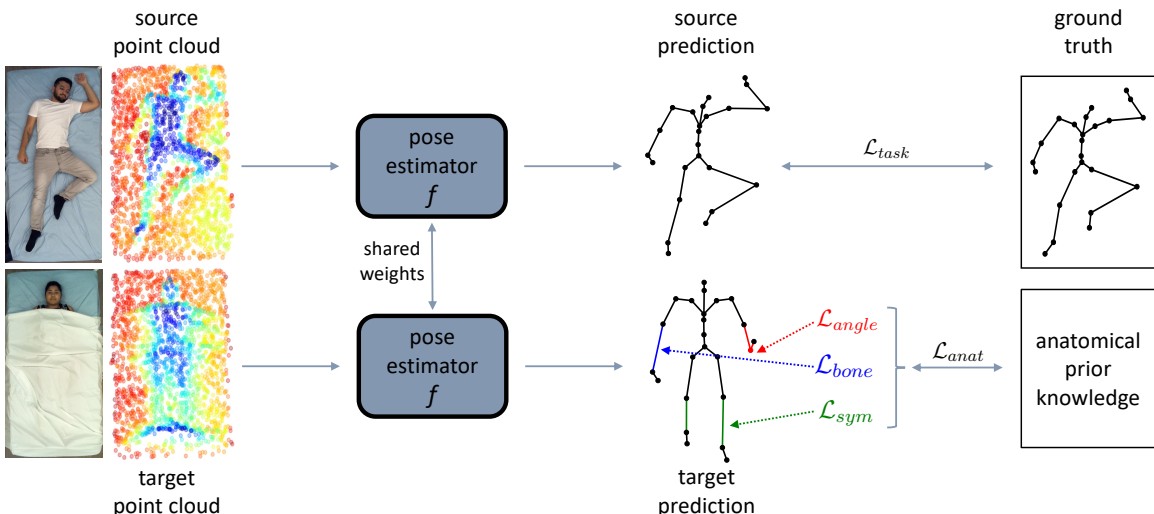

Figure 1: Overview of our method. While minimizing a supervised task loss on source data, we jointly constrain target predictions to match prior knowledge about human anatomy. This is implemented by an anatomical loss that penalizes asymmetric limb lengths, implausible bone lengths and implausible joint angles. Color images are not used in our framework and are only shown for better visualization.

loss and supervises the learning process in the labeled source domain. The anatomical loss is computed on target data and penalizes implausible predictions that violate certain anatomical constraints, provided in the form of prior knowledge about human anatomy. That way, the anatomical loss guides the learning process in the target domain by weak supervision in the output space and encourages the learning of meaningful task-relevant features in this domain. Thus, the anatomical loss is the crucial domain-adaptive component of our method and its careful design is critical.

## 3.1. Weak supervision through anatomical constraints

The anatomical loss is supposed to penalize implausible predictions. The essential question is how to measure anatomical plausibility or—in other words—what form of prior anatomical knowledge to embed in the loss function. Considering human joints as the nodes of the human skeleton graph, we identify three measurable properties of the skeleton that can be associated with anatomical plausibility and readily be embedded in a loss function.

1. Human limbs usually have symmetric lengths. Therefore, we penalize predictions with asymmetric limb lengths by a symmetry loss $\mathcal{L}_{sym}$. Let $\mathcal{B} = \{\boldsymbol{b}_i\}_{i=1}^{N_\beta}$ denote the set of all bone vectors $\boldsymbol{b}_i \in \mathbb{R}^3$ that connect two joints in a predicted skeleton graph $\hat{\boldsymbol{Y}}$. Let further $\mathcal{B}_\lambda \subset \mathcal{B}$ denote the subset of $N_\lambda$ bones $\boldsymbol{b}_i^\lambda$ of the left body side that have a counterpart $\boldsymbol{b}_i^\rho \in \mathcal{B}_\rho$ on the right body side. In practice, that includes arms and legs. The symmetry loss is then defined as

$$\mathcal{L}_{sym} = \frac{1}{N_\lambda} \sum_{i=1}^{N_\lambda} \left| \|\boldsymbol{b}_i^\lambda\|_2 - \|\boldsymbol{b}_i^\rho\|_2 \right| \qquad (2)$$

2. Bones $\boldsymbol{b}_i$ of the human body have typical lengths, which can be constrained by bone-specific upper and lower bounds $u_i^\beta$ and $l_i^\beta$. Predictions with bone lengths outside this range are penalized by the bone loss

$$\mathcal{L}_{bone} = \frac{1}{N_\beta} \sum_{i=1}^{N_\beta} \ell(\|\boldsymbol{b}_i\|_2; l_i^\beta, u_i^\beta) \quad \text{with} \quad \ell(x; l, u) = \begin{cases} |x - l| & x < l \\ |x - u| & x > u \\ 0 & l < x < u \end{cases} \tag{3}$$

Here, $u_i^\beta$ and $l_i^\beta$ can be inferred from the training set or an anatomical textbook.

3. Human joints cannot freely rotate by 360 degrees but have a joint-specific limited range of angles that can be taken. In other words, the scalar product between two (normalized) bone vectors $\boldsymbol{b}_i$, $\boldsymbol{b}_j$ that are connected by a joint is constrained by upper and lower bounds $u_{ij}^\alpha$ and $l_{ij}^\alpha$. We impose this constraint by minimizing an angle loss $\mathcal{L}_{angle}$. Let $\mathcal{B}_\zeta = \{(\boldsymbol{b}_i, \boldsymbol{b}_j)\}$ be the set of all $N_\zeta$ pairs of bone vectors that are connected by a joint. We then define the angle loss as

$$\mathcal{L}_{angle} = \frac{1}{N_\zeta} \sum_{(\boldsymbol{b}_i, \boldsymbol{b}_j) \in \mathcal{B}_\zeta} \ell\left(\frac{\boldsymbol{b}_i}{\|\boldsymbol{b}_i\|_2} \cdot \frac{\boldsymbol{b}_j}{\|\boldsymbol{b}_j\|_2}; l_{ij}^\alpha, u_{ij}^\alpha\right) \tag{4}$$

with $\ell(x; l, u)$ as in (3). Again, upper and lower bounds $u_{ij}^\alpha$ and $l_{ij}^\alpha$ can be inferred from the training set or an anatomical textbook.

Altogether, we define the anatomical loss function as

$$\mathcal{L}_{anat} = \mathcal{L}_{sym} + \mathcal{L}_{bone} + \mathcal{L}_{angle} \tag{5}$$

### 3.2. Optimization

When minimizing the overall loss (1) over all model parameters jointly, we observed that the model tends to learn two distinct functions that separately minimize the loss functions of the two domains. This lead to a mode collapse in the target domain, where the model predicted an anatomically plausible but input-independent fixed pose. To prevent this, we reduce the effective model capacity during optimization in the target domain and focus on learning the feature extractor. We split the function $f$ in a feature extractor $g$ and network heads $h$ (both shared among domains), i.e. $f(\boldsymbol{X}; \boldsymbol{\theta}_f) = h(g(\boldsymbol{X}; \boldsymbol{\theta}_g); \boldsymbol{\theta}_h)$, and minimize the anatomical loss on the target domain only with respect to $\boldsymbol{\theta}_g$ while keeping $\boldsymbol{\theta}_h$ fixed:

$$\boldsymbol{\theta}_g^* = \min_{\boldsymbol{\theta}_g} \mathcal{L}_{task} + \lambda \mathcal{L}_{anat}, \quad \boldsymbol{\theta}_h^* = \min_{\boldsymbol{\theta}_h} \mathcal{L}_{task} \tag{6}$$

### 3.3. Point cloud-based 3D pose estimation

While our formulation is agnostic to the specific implementation of the function $f$, we realize point cloud-based 3D pose estimation as follows. Given an input point cloud $\boldsymbol{X} \in \mathbb{R}^{N \times 3}$, we estimate the associated 3D pose $\hat{\boldsymbol{Y}} \in \mathbb{R}^{K \times 3}$ as a weighted sum over the $N$ input points $\boldsymbol{x}_i \in \mathbb{R}^3$. For this purpose, we design $f$ to output a stack of $K$ softmax-normalized weight maps $\boldsymbol{W} = f(\boldsymbol{X}; \boldsymbol{\theta_f}) \in \mathbb{R}^{N \times K}$ over the input points. The $k$-th predicted joint is then given by $\hat{\boldsymbol{y}}_k = \sum_{i=1}^N \boldsymbol{x}_i \cdot w_{ik}$. In our work, we implement $f$ as the segmentation architecture of DGCNN (Wang et al., 2019) with 40 neighbors in the knn-graph. We split the network into $g$ and $h$ after the last encoding layer (conv6).

## 4. Experimental setup

**Dataset.** We evaluate our method on the SLP dataset (Liu and Ostadabbas, 2019; Liu et al., 2020), which shows human subjects lying in bed, simulating the use case of patient monitoring. The dataset comprises single-view depth frames of 109 subjects. Each subject takes 45 poses in supine and lateral (left, right) positions. For each pose, the subjects do not move until three frames with varying cover conditions (no cover, thin cover $\sim$1 mm, thick cover $\sim$3 mm) are taken. That way, ground truth poses annotated on frames without a cover are also valid for frames with cover. While the original dataset includes 2D joints, Clever et al. (2021) provided the 24 joints of the SMPL model (Loper et al., 2015) as 3D ground truth for the first 102 subjects. We restrict our experiments to these subjects. The first 70 subjects are used for training, subjects 71-80 for validation, and subjects 81-102 for testing. As a pre-processing step, we transform depth frames to point clouds. To this end, we first use depth thresholding to detect the pixels belonging to patient and bed and subsequently lift these pixels to 3D space using the internal camera parameters.

**Adaptation scenario.** We consider uncovered subjects as the labeled source domain and covered subjects as the unlabeled target domain. Thus, the domain shift consists in the occlusion of the subject by a cover. The scenario is relevant in practical applications because the annotation of uncovered subjects is viable while it is infeasible for covered patients in practice. For our experiments, we randomly divide the training data by subject into three splits with 30, 20, and 20 subjects. For each split, we use only one cover condition—uncover, thin cover, and thick cover, respectively—while the remaining data is discarded. This yields 30 subjects as the source domain and 40 subjects as the target domain. For validation and test set, we use both the thin and the thick cover for all frames of all subjects. Final results are reported on the test set in form of the mean per joint position error (MPJPE).

**Implementation details.** We implement our framework in PyTorch and optimize model parameters with the Adam optimizer. To prevent noisy gradients from $\mathcal{L}_{anat}$ at early epochs, we pretrain our model on source data for 15 epochs with a learning rate of 0.001. Next, using mixed batches of half source and half target data, we optimize for the joint loss function (1) with $\lambda = 0.1$ for 100 epochs with an initial learning rate of 0.001, which is divided by 10 at epochs 60 and 90. For regularization, we use a weight decay of 1e-5 and augment input point clouds by random translation, rotation and subsampling to 2048 points. Upper/lower bounds $u_{ij}^{\alpha}$, $u_{i}^{\beta}$ / $l_{ij}^{\alpha}$, $l_{i}^{\beta}$ are set to the max/min values from the training set.

**Baselines.** As lower and upper bound, we train our model on labeled source data and labeled target data, respectively, without any adaptation techniques. Moreover, we adapt diverse state-of-the-art domain adaptation techniques. 1) From the area of domain-invariant feature learning, we select *MMD* (Tzeng et al., 2014) and *DANN* (Ganin and Lempitsky, 2015) and apply them to the global feature vector after conv6 in the DGCNN. 2) Domain adaptation through self-supervision: a) We adapt deformation-reconstruction (*DefRec*) by Achituve et al. (2021). b) We predict the *displacement* vector between two sampled patches of the input cloud, which is similar to the auxiliary task proposed by Doersch et al. (2015). 3) From the field of *self-training with noisy pseudo labels*, we apply consistency-constrained semi-supervised learning by Mu et al. (2020). 4) As for *self-ensembling*, we adapt the teacher-student approach by French et al. (2017). 5) We adapt the optimization strategy of

maximum classifier discrepancy (*MCD*) by Saito et al. (2018). Since step B of their method lead to divergence in our case, we discarded this step. 6) We realize *adversarial output space adaptation* (Yang et al., 2018) by training a discriminator to distinguish predicted skeletons in the target domain from ground truth skeletons in the source domain. Technically, this baseline is closest to our method, and the discriminator could—in theory—learn to penalize implausible predictions in a similar way as our anatomical loss. For all baseline models and our method, hyper-parameters are optimized on the validation set of the target domain.

## 5. Results

Quantitative results of our experiments are shown in Tab. 1, and qualitative results are presented in App. A.1. First, we note that the source-only baseline[1] performs clearly worse than the target-only model, increasing the MPJPE by 93 %. This underlines the severity of the domain gap and confirms the need for effective adaptation techniques to close the gap.

Second, the results show that our method successfully addresses the problem. Each of the loss functions $\mathcal{L}_{sym}$, $\mathcal{L}_{angle}$ and $\mathcal{L}_{bone}$ alone already reduces the mean error from 130.4 mm to 105.9 mm, 106.7 mm and 102.9 mm, respectively. Aggregating them in the joint loss $\mathcal{L}_{anat}$ further reduces the error to 96.6 mm. Overall, this corresponds to a relative improvement of 26 % while the gap between source-only and target-only model is reduced by 54 %. Regarding specific joints, the improvement by our method is particularly notable for foot, knee, elbow, and hand joints, amounting to 32.6, 45.9, 55.7, and 84.8 mm, respectively.

Third, we compare our method to state-of-the-art domain adaptation techniques. Our method outperforms all competing methods and achieves the lowest average error. The improvement over all competitors is statistically significant ($p < 0.01$) as confirmed by a Wilcoxon signed-rank test. Notably, our method surpasses adversarial output adaptation, highlighting the efficacy of explicit constraints as opposed to adversarial optimization.

In an additional experiment, we combine our method with the best competing approaches, namely MCD, self-ensembling, DANN, and self-training—see App. A.2, Tab. 2 for detailed results. These combinations achieve an MPJPE of 95.7 mm, 92.3 mm, 95.1 mm, and 94.5 mm, respectively, and thus consistently surpass the performance of the individual methods. This demonstrates the versatility of our approach. Finally, we provide an ablation study on the choice of loss functions (2), (3), and (4) in App. A.3.

## 6. Conclusion

We tackled domain adaptation for 3D human pose estimation by imposing anatomical constraints on target predictions. Our experiments showed that our anatomical loss function effectively guides the learning process in the target domain and constitutes a powerful form of weak supervision in the absence of labels. For patient monitoring on the SLP dataset, our method surpassed diverse competing methods while favoring anatomically plausible pose estimates. Quantitatively, our work improved the mean error of pose estimates by 26 % from 13 cm to less than 10 cm, which can advance the reliability of clinical monitoring systems.

While these are promising results, they apply to healthy subjects fulfilling our anatomical constraints. By contrast, patients in the clinic may violate the constraints due to

---

1. The source-only model is already far better than using a mean pose as an estimate (MPJPE=185.8 mm).

Table 1: Results for uncover→cover adaptation on the SLP dataset. We compare the MPJPE [mm] of our method to the baselines. Results are averaged over thin and thick cover as the scores are almost identical.

| Method | Feet | Knees | Hips | Core | Head | Shoul | Elb | Hands | Mean |
|---|---|---|---|---|---|---|---|---|---|
| source-only | 174.1 | 148.1 | 74.5 | 56.5 | 34.8 | 65.7 | 168.2 | 273.2 | 130.4 |
| target-only | 86.4 | 64.8 | 36.7 | 31.6 | 29.4 | 42.3 | 80.6 | 140.0 | 67.7 |
| MMD | 164.6 | 124.6 | 68.5 | 56.9 | 35.3 | 62.8 | 177.1 | 243.0 | 121.7 |
| DANN | 168.8 | 114.5 | 60.9 | 50.3 | 33.3 | 55.0 | 144.8 | 218.8 | 111.6 |
| DefRec | 161.0 | 130.6 | 68.1 | 51.4 | 34.5 | 63.6 | 175.3 | 255.0 | 122.6 |
| displacement | 168.4 | 122.7 | 65.7 | 51.0 | 33.9 | 59.9 | 165.1 | 258.4 | 121.9 |
| output adapt. | 181.4 | 128.6 | 62.9 | **47.1** | 35.5 | 59.3 | 136.8 | 207.9 | 112.9 |
| self-training | 144.9 | 134.1 | 71.7 | 54.9 | 33.6 | 59.7 | 145.4 | 222.3 | 112.4 |
| MCD | 151.8 | 116.8 | 63.7 | 52.6 | 33.6 | 53.1 | 120.4 | **171.4** | 99.4 |
| self-ensembling | 155.9 | 109.8 | 73.6 | 57.4 | 35.0 | 56.1 | 118.6 | 175.9 | 102.3 |
| ours, $\mathcal{L}_{sym}$ only | 148.6 | 116.1 | 66.2 | 53.1 | 33.4 | 53.6 | 140.7 | 201.7 | 105.9 |
| ours, $\mathcal{L}_{angle}$ only | 155.9 | 118.2 | 64.3 | 52.2 | 34.5 | 58.5 | 134.3 | 198.1 | 106.7 |
| ours, $\mathcal{L}_{bone}$ only | 144.3 | 107.9 | 60.5 | 51.0 | **32.4** | 52.2 | 128.9 | 205.0 | 102.9 |
| ours | **141.5** | **102.2** | **56.0** | 47.2 | 33.3 | **50.4** | **112.5** | 188.4 | **96.6** |

pathological abnormalities (asymmetric/deformed limbs) or extreme body dimensions. The used symmetry and bone losses could severely impair pose estimates of such patients. However, our method offers sufficient flexibility to prevent such problems by carefully adapting the constraints. The hard symmetry constraint can be relaxed to a soft inequality constraint ($|\|\boldsymbol{b}_i^\lambda\|_2 - \|\boldsymbol{b}_i^\rho\|_2| < \delta$), and the bounds of the bone loss can be set according to the expected target population. Another open clinical problem is the detection of missing limbs, which could either be approached in an uncertainty-driven manner or by formulating pose estimation as an object detection problem (McNally et al., 2021). Finally, clinical settings include domain shifts beyond the treated occlusion problem (e.g. a different bed or a varying camera perspective). While the formulation of our method is agnostic to the specific shift, its effectiveness under such settings needs to be verified in future experiments.

As a methodical outlook, we believe that the potential of anatomical priors is not fully exploited yet. First, our formulation of the angle loss still permits implausible poses because 1) joints are considered in isolation, 2) the scalar product cannot uniquely represent the space of 3D rotations. The incorporation of a kinematic model could overcome these shortcomings. Second, instead of providing prior anatomical knowledge in form of a loss, the underlying constraints could be embedded into the network architecture, preventing implausible predictions by design. This could improve model robustness and domain generalization. Third, in practice, the approximate bone lengths of a subject might be a priori known at test time (e.g. from a previous highly confident estimate). While this should simplify the pose estimation, it is an open question of how to exploit this knowledge in an uncertainty-aware manner. In summary, our work thus demonstrates the merit of efficiently applied anatomical prior knowledge and opens promising directions for future work. Finally, beyond human pose estimation, our method could be adapted to general anatomical landmark detection, which is of interest for medical imaging.

## Acknowledgments

We gratefully acknowledge the financial support by the Federal Ministry for Economic Affairs and Energy of Germany (FKZ: 01MK20012B).

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

## Appendix A. Experimental results

### A.1. Qualitative results

Qualitative results of our main experiment are shown in Fig. 2. Predictions by our model are visually more accurate and appear anatomically more plausible. Specifically, our anatomical constraints prevent implausible bone lengths of lower and upper arm (all rows) and lower and upper leg (first and third row) as well as implausible angles in shoulder, elbow, and wrist joints (first, second, and fourth row). A failure case is shown in the last row, where the prediction appears anatomically plausible but is inconsistent with the actual pose.

### A.2. Combination of our method with the state of the art

In this experiment, we combine our method with the best competing approaches from Tab. 1, namely MCD, self-ensembling, DANN, and self-training. Detailed results of this experiment are presented in Tab. 2. For each of the four methods, the combination with our method surpasses the performance of the method itself as well as the performance of our method.

Table 2: Performance comparison for the combination of our method with the best competing approaches. Results are reported in terms of MPJPE [mm] for uncover→cover adaptation on the SLP dataset.

| Method | Feet | Knees | Hips | Core | Head | Shoul | Elb | Hands | Mean |
|---|---|---|---|---|---|---|---|---|---|
| source-only | 174.1 | 148.1 | 74.5 | 56.5 | 34.8 | 65.7 | 168.2 | 273.2 | 130.4 |
| target-only | 86.4 | 64.8 | 36.7 | 31.6 | 29.4 | 42.3 | 80.6 | 140.0 | 67.7 |
| ours | 141.5 | 102.2 | 56.0 | 47.2 | 33.3 | 50.4 | 112.5 | 188.4 | 96.6 |
| DANN | 168.8 | 114.5 | 60.9 | 50.3 | 33.3 | 55.0 | 144.8 | 218.8 | 111.6 |
| DANN+ours | 136.5 | **98.1** | 57.8 | 48.8 | 33.5 | **50.0** | 113.1 | 183.9 | 95.1 |
| self-training | 144.9 | 134.1 | 71.7 | 54.9 | 33.6 | 59.7 | 145.4 | 222.3 | 112.4 |
| self-train.+ours | 137.0 | 101.1 | **55.6** | 48.2 | **33.1** | 50.2 | 110.3 | 181.7 | 94.5 |
| MCD | 151.8 | 116.8 | 63.7 | 52.6 | 33.6 | 53.1 | 120.4 | 171.4 | 99.4 |
| MCD+ours | 139.3 | 104.8 | 59.5 | 50.0 | **33.1** | 52.1 | **110.1** | 179.1 | 95.7 |
| self-ensembling | 155.9 | 109.8 | 73.6 | 57.4 | 35.0 | 56.1 | 118.6 | 175.9 | 102.3 |
| self-ensemb.+ours | **135.6** | 104.1 | 57.5 | **47.1** | 34.3 | 52.1 | 110.4 | **165.7** | **92.3** |

### A.3. Ablation study: loss functions

In this ablation experiment, we examine the optimal choice of loss functions for the symmetry constraint (2), the bone length constraint (3), and the angle constraint (4). To this end, we train our method with each of the three constraints separately and compare the effect of a linear L1 penalty (as used by our method) and a quadratic L2 penalty. Training and test setup is identical to our main experiment. Results are presented in Tab. 3 and

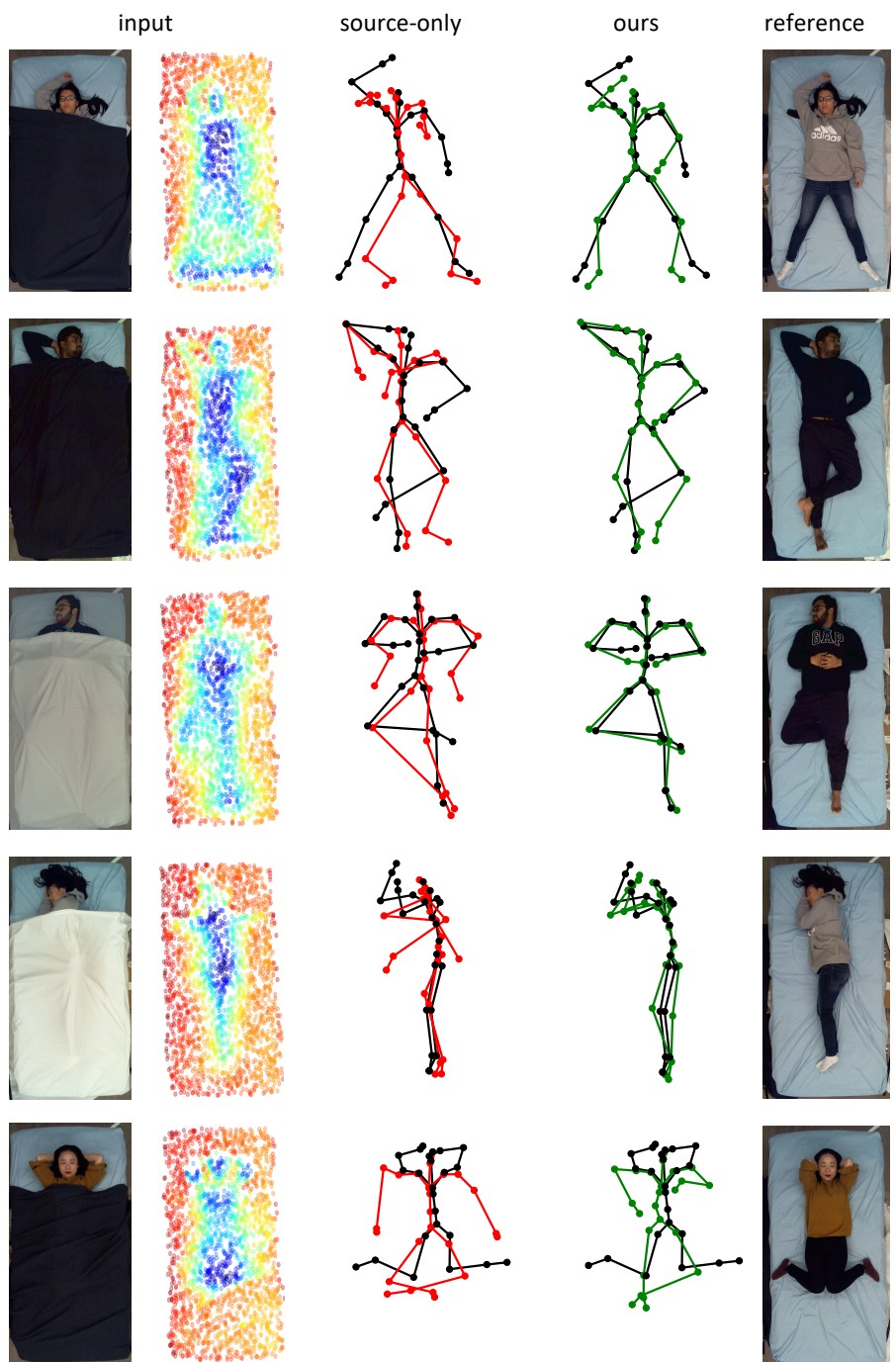

Figure 2: Qualitative results on five samples from the SLP dataset. We show predictions by the source-only model (red) and by our model (green) together with the ground truth (black). Input point clouds are shown together with the associated RGB images for better visualization. The corresponding RGB image without a cover is given for reference.

Table 3: Results of the ablation experiment on different loss functions. For each of the three anatomical constraints, we compare a linear L1 loss against a quadratic L2 loss. Evaluation was performed under the uncover→cover adaptation scenario on the SLP dataset.

| Method | L1 | L2 | MPJPE [mm] |
|---|---|---|---|
| $\mathcal{L}_{sym}$ only | ✓ | | **105.9** |
| $\mathcal{L}_{sym}$ only | | ✓ | 108.6 |
| $\mathcal{L}_{angle}$ only | ✓ | | **106.7** |
| $\mathcal{L}_{angle}$ only | | ✓ | 119.3 |
| $\mathcal{L}_{bone}$ only | ✓ | | **102.9** |
| $\mathcal{L}_{bone}$ only | | ✓ | 104.1 |

show that an L1 loss yields a better performance for all three constraints, whereby the gap is particularly remarkable for the angle loss.

