# OpenReview forum: "Domain adaptation through anatomical constraints for 3d human pose estimation under the cover"
_MIDL.io/2022/Conference — MIDL 2022_

### Official Review · Reviewer_jcjL · 2022-01-21

**Confidence:** 4
**Preliminary Rating:** 5
**Recommendation:** Oral, Poster

**Summary:**

The paper addresses domain transfer for human pose estimation. The scenario is simple: there are only pose datasets with humans without a cover, we want to generalize to subjects that have a blanket covering them. We do not have ground truth poses for this second scenario. In this context, the authors introduce additional anatomically motivated losses to regularize the network outputs with "plausible" poses, thus introducing domain (anatomic) knowledge (even if explicit values are again derived from the dataset). The paper has a strong baseline set of 8 methods and achieves superior performance overall and across 6 of 8 individual "end points", i.e., the method outperforms the state-of-the-art. The errors significantly increase with distance from the core and "size" of the target point, which is plausible, as this information will be hard to uncover from the data.

**Strengths:**

1. The paper follows best-practice for dataset hygiene and splits. Dedicated and plausible decisions are made for training/test/validation and even domain splits. I wish every paper was following this basic best-practice. You can even add that dividing the training data into split for the different domains is necessary to reduce confounders (otherwise could cause overfitting to unrelated features).
2. In total 8 different baselines allow for a thourough evaluation and comparison of the method. Very much appreciated!
3. The paper outperforms the state-of-the-art clearly.
4. Even Results of combinations of the proposed loss and baseline method are reported, however not included in the Table 1.
5. Very well written and structured, easy to read.

**Weaknesses:**

Major
1. The impact of the work is probably somewhat limited. At least for me it is unclear, how this would translate to other applications. The authors suggest landmark detection as a more general scenario for its application, but I am not aware of any landmark detection secario that uses point clouds. So I believe the solution is mostly limited to this specific problem.
2. The problem seems plaugued by limited information. The authors even experienced difficulty to estimate deviations from the mean pose in the initial training. I would like to see a new baseline added to understand the "quality". The trivial baseline: the mean pose. In my mind, this is equivalent to a coin-flip in classification and helps understand whether this approach is actually going anywhere. I would anchor the mean pose on head and core. If the error of the source training (worst-case) is less than this mean pose approach, a simple comment suffices.

Minor
1. I don't like the naming
$\cal{L}_{bone}$,
I'd prefer this to be something like
$\cal{L}_{length}$
 this causes a bunch of little changes throughout the paper. [sorry openreview is bad at multiple equations per line :( ]
2. Results of combinations of the losses with the baselines are not reported in Table 1. It is unclear to me, why results are reported in the text, but not in the Table -- or even an additional Table in the appendix if space-constraints were the issue.
3. Discussion of Limitations:
-  What happens, when the subject violates the anatomical constraints? Do you expect any performance difference in methods then? Even is there a performance difference within the dataset? Examples: amputated/deformed limbs, children/very small humans
- Does the symmetric limb loss introduce a bias towards people, that actually do have asymmetric limbs (as part of a pathelogical condition)? Likely, this will not be covered in the dataset, but from a clinical standpoint, this is relevant.
- What are the limitations of the approach?
- I would expect different priors for patients covered with a blanket and without a blanket, at least this should be disscussed.
4. page 6 para 2: The statement of "challenging setting ... for human observers" requires qualifiers: 1. based on depth images? or RGB? 2. evidence (appendix) or add "in your experience" or citation.
5. page 6, para 1 last sentence: I find this a bit unclear. What happens here? how does this processing work?

**Deanonymize Review:**

no

**Detailed Comments:**

1. Why are those hard limits (bone and angle loss) good? Would a statistical distribution loss (e.g. KL-divergence on the observed distribution) not be a better idea?
2. Loss Normalization: in your formulation, there is a fixed factor for the loss, but the magnitude of the loss might be very different, specifically, for angles vs. lengths. Maybe losses should be normalized to account for this phenomenon of different magnitudes.
3. Your architecture already is an Encoder-Decoder design. If you wanted to ensure the network is more constrained to "plausible" poses, you could randomly sample the latent space and apply your plausability constraints on the "random poses". I have actually implemented such designs for encoder-decoder and auto-encoder designs and achieved good results with that.

**Final Rating After The Rebuttal:**

5: Strong Accept

**Justification Of The Final Rating:**

This work is a thorough analysis of th problem of pose estimation under occlusion.
The discussion of limitations has added to the papers depth.
Had hoped the authors would also have included the comment about the mean pose (somewhere) in the paper, because it is interesting context information for anyone reading the work (even if it is just a comment in the text like "source is already better than averaged joint positions")
Despite the comment, I still dont quite see more applications, but this does not really limit my evaluation.

I am not able anymore to adjust my "Special Issue" Recommendation, but an extended version seems like a good fit for a Special Issue in my eyes.

**Paper Type:**

methodological development

**Questions To Address In The Rebuttal:**

Well, this is basically the weaknesses and the detailed comments. I think they are already structured in a way that indicates how the questions should be addressed.
In "Detailed comments", I list some comments for future work and maybe a discussion, which are not really needed to be addressed in the paper.

The two "major weaknesses" and the "ignorance" w.r.t. limitations (i.e. a discussion of limitations) are holding me back from a more positive evaluation (Oral/Best Paper/Special Issue).

**Special Issue:**

no

---

### Official Review · Reviewer_dGNs · 2022-01-26

**Confidence:** 4
**Preliminary Rating:** 5
**Recommendation:** Best Paper Award, Oral, Poster

**Summary:**

This paper tackles the problem of 3D pose estimation in bed-ridden patients, with a 3D point cloud as an input. One of the main difficulty of the task is the impossibility to directly annotate data, due to bed-covers.
What the authors do is to resort to domain adaptation, using as a source a specially crafted dataset with and without cover (so that the pose is known and available), and then targetting (fine-tuning) a set with a cover (and without the pose available). As such, we go from the "no cover" domain, to the "cover" domain.

The novelty here is to regularize the fine-tuning using anatomical priors (symmetry, limb size and joints angles), which are encoded as series of constraints into the combined loss function. This improves the results by penalizing unrealistic predictions. Such constraints are not needed as inference time, this is only done during the fine-tuning stage.

The framework presented is very simple and elegant, and can be easily extended to encode other priors, and/or corner cases.


---
I do not see a field for that this year, so I'll state it here: I am not familiar with the pose estimation litterature, and if the authors had decided to ignore a whole class of papers I would not be aware of it. Hence the '4' on my own confidence score.

**Strengths:**

- The paper is very well written and well motivated.
- Simple and elegant framework
- can be extended with other priors, and/or plugged into existing methods as additional regularization
- compare to numerous other domain adaptation methods (this should not be down-played, as it is often the most time-consuming task)
- The experimental protocol is sound

**Weaknesses:**

- The training regiment could be described in more details. At the moment, it is not clear how exactly the two sets (target and source) are used.


No more weaknesses, but I have to fill the 200 chars count.

**Deanonymize Review:**

yes

**Detailed Comments:**

While the paper is very well explained and motivated overall, I find that Section 3 falls into the (somewhat common) caveat of presenting the method, without necessarily motivating the details. My personal take would be to first present it as a constrained optimization problem, and _then_ mention how they are embedding at training (slightly tuned alphabet):

$\\begin{align}
\\min_{\\theta} \\sum_{s \\in \\mathcal S} & \\sum_{k \\in \\mathcal K} ||\\hat y_s^{(k)} - y_s^{(k)}||_1 \\\\
\\text{s.t. } & ||b^{(r)}||_2 = ||b^{(l)}||_2 & \\forall (b^{(r)}, b^{(l)}) \\in \\mathcal B_l^t, \\forall t \\in \\mathcal T \\\\
        & l_i \\leq ||b\_{t, i}|| \\leq {u_i} & \\forall i \\in \\{1, ..., N_b\\}, \\forall t \\in \\mathcal T \\\\
        & l^{(\\alpha)}\_{ij} \\leq \\frac{b_i}{||b_i||_2} . \\frac{b_j}{||b_j||_2} \\leq u^{(\\alpha)}\_{ij} & \\forall ij \\in \\mathcal B^t\_{\\text{conn.}}, \\forall t \\in \\mathcal T
\\end{align}$

(With $\mathcal S$ the source set, $\mathcal T$ the target set, and the rest should be sufficiently close to the initial notation to be understandable.)

It would then be easier to discuss (and motivate):
- why a quadratic penalty is used for the equality constraint (Eq (2)), while a linear penalty is used for the inequality constraints (Eq (3) and (4));
- would make it easier to relax the symmetry equality for patients with non-symmetric limbs;
- as you discuss very interesting other priors in the conclusion (the overall consistency across all joints), I suspect that optimizing that function might require more powerful optimization methods, so decoupling the model from the optimization might help;
- how the regularization could easily be applied to the source set as well;
- highlights more how the bounds for Eq. (3) are patient-independent, but could easily embed other information (as you mention in the conclusion).

---
Minor:
- few notation clashes, for instance the $l$ of $N_l$ is re-used for $l_i^b$; $b$ in $N_b$ and $u_i^b$ reads a bit like an index. It slightly hamper the reading, and could easily be avoided by using other alphabets (e.g. greek) or fancy formatting (mathcal, mathbb, mathfrak)
- A Table 2 containing the results from the last paragraph of 4.4 is needed, I almost missed those (really interesting) results
- Reminding in 4.2 that your code is available online can be useful, for readers that haven't checked the abstract yet or simply forgot about it
- page 4: should either be "or -- in other words -- what" or "or---in other words---what" (en or em-dash, not a single dash)


**Final Rating After The Rebuttal:**

5: Strong Accept

**Justification Of The Final Rating:**

The authors did not screw up their rebuttal, so my strong accept remains a strong accept. Congratulation for the great paper!

I also thank (and commend) the authors for their thorough responses too all questions and minor concerns from the reviewers.

---
> We thank the reviewer for the proposed alternative formulation and appreciate its advantages. Due to space constraints, the formulation does not fit in our methods section, but it is a valuable input for future work.

To explicit things, a lot of my suggestions were for a future (special issue) extension; I believe that this work has a lot of potential and interesting ideas to explore.

**Paper Type:**

methodological development

**Questions To Address In The Rebuttal:**

### On the formulation
- Why is the symmetry handled with a quadratic penalty (Eq. (2)), while the limb length and angles are handled (Eq. (3), (4)) are handled with a linear penalty? Do you have theoritical or empirical reasons for that?
- For the limb length, would it be possible to refine the range $[l^a_{ij}, u^a_{ij}]$? For instance by taking into account the patient height ("scale")? Or is there too much variability at play? Perhaps a simplified version could already help, such as the $\frac{\text{femur}}{\text{humerus}}$ ratio
- Would the same regularization help on the source training? What are the mistakes done by the network at that stage?

How the training is performed? With both the source and target sets, I see at least several options:
- two stage: train on source and fine-tune on source;
- alternate continuously over the two until convergence;
- train with the source for one epoch, then perform mixed batches (half source, half target).

### On the network architecture
I see that the network encodes the $K$ joints as a classification problem over the $N$ cloud points. This raises a few interrogations to me:
- has it been investigated (in the litterature) to perform it as a regression problem? (i.e. $f: \mathbb R^{N\times3} \rightarrow \mathbb R^{K\times3}$)
- how could it handle patients with, e.g., missing limbs? Would you just mask parts of the output manually, or could you devise a way for the network to predict that some joints are missing.

### Misc.
- I see that the patiens ID are not shuffled before split for the train/val/test sets. Is there any chance that the original dataset sorted the patients in some way? This could bias the results.

**Special Issue:**

yes

---

### Official Review · Reviewer_cbbY · 2022-01-26

**Confidence:** 4
**Preliminary Rating:** 3
**Recommendation:** Poster

**Summary:**

The paper introduces anatomically constrained losses in 3D human pose estimation; and joint training with two groups of datasets where one of the groups has annotated labels of the joints providing supervisory signal (for “task loss”) and the other one uses the anatomical constrained loss penalizing asymmetric limb lengths, implausible bone lengths, and implausible joint angles. The authors pose this as a domain adaptation where the source domain is the set of point clouds obtained from the first group of images of humans lying in the bed, and the target domain point clouds obtained from the second group of images of humans lying in the bed but occluded by the blanket cover of varying thickness. The comparison of the proposed method against various other methods show that the proposed method can provide better results.

**Strengths:**

Introducing anatomical constraints in losses and using joint training to learn a common feature extractor is a simpler approach but looks quite effective compared to adversarial regularizer.

The method is compared against several other state-of-the-art methods and the results of show that the proposed approach is quite effective for this particular dataset and setup.

**Weaknesses:**

The title and the positioning of the paper as domain adaptation seem to be a bit too broad than the setup where the paper focuses on images of a person covered and not covered in blankets. While this is technically “domain adaptation” for a very specific source domain and target domain type, it is not clear how much of the benefit this method brings to other kinds of domain shifts, especially population shift (age, pathology etc.) where the anatomical prior obtained from the source domain may not work well in the target domain.

If other forms of domain shift where there is no occlusion are considered, it is not clear how much benefit the proposed method brings compared to other domain adaptation methods because the input data would have more information about the joints than the occluded scenario. The paper does not explore this in the experiments or at least discuss it in the discussion.

**Deanonymize Review:**

no

**Detailed Comments:**

One of the motivations for using point clouds is mentioned to be privacy reasons. However, it is not clear how this translates in the practical use case where estimating point cloud step needs to use the original images.

The paper mentions that the prior knowledge about human anatomy in the unlabeled scenario has not been fully exploited. Perhaps, the positioning of the work may be improved by adding literature on anatomical constraints with template matching that is common in medical imaging. It has been explored for human pose estimation as well [A]

Sec 4.1 train-test split has ensured no leak w.r.t patients but it seems like leak w.r.t to the pose is not considered. It would be interesting to see how well the method performs when test images contain poses that are not seen at all in training.

Sec 3.2 explanation about splitting the neural network into feature extractor g and network heads h is a bit confusing. Are there two heads or only one head for the source domain while using g for the target domain? I think f(h(g)) is used for both but h is not updated with anatomical loss. A figure showing g and h would be helpful.

Minor typos:

Sec 3.1 “Let further denote …" Grammar

Just before Eq (6), fix -> fixed

Schmidtke, Luca, et al. "Unsupervised Human Pose Estimation through Transforming Shape Templates." Proceedings of the IEEE/CVF Conference on Computer Vision and Pattern Recognition. 2021.

**Final Rating After The Rebuttal:**

4: Weak Accept

**Justification Of The Final Rating:**

The authors have sufficiently addressed my concerns in their responses with regards to the positioning of the paper and the discussion on the limitations. The proposed work will be of interest to the MIDL community, hence upgrading the rating.

**Paper Type:**

both

**Questions To Address In The Rebuttal:**

I would like to know how the paper could make clearer the scope of the “domain-adaptation” in this paper and discuss potential limitations of the proposed method for domain shifts other than the occlusion.

**Special Issue:**

no

---

### Meta-Review · Area_Chair_bmTB · 2022-02-19

**Recommendation:** Accept (Oral)
**Confidence:** 5

**Metareview:**

This paper introduces anatomically constrained losses for 3D human pose estimation, in a domain-adaptation setting. The three reviewers agree that this is an excellent and well-executed work, and I concur with this. The method is simple/elegant/well-motivated, and the comprehensive experiments/comparisons show clearly the usefulness of embedding this type of prior-knowledge constraints as loss functions. The paper is well-written, and the authors provided a detailed answer to the reviews, including additional ablation studies.

---

### Decision · Program_Chairs · 2022-02-28

Accept